# Revisiting the Role of Serotonin in Sleep-Disordered Breathing

**DOI:** 10.3390/ijms25031483

**Published:** 2024-01-25

**Authors:** O Aung, Mateus R. Amorim, David Mendelowitz, Vsevolod Y. Polotsky

**Affiliations:** 1Department of Medicine, Johns Hopkins University, Baltimore, MD 21224, USA; oaung@mcw.edu (O.A.); mateus.ramosamorim@gwu.edu (M.R.A.); 2Medical College of Wisconsin, Milwaukee, WI 53226, USA; 3Department of Anesthesiology and Critical Care Medicine, George Washington University, Washington, DC 20037, USA; 4Department of Pharmacology and Physiology, George Washington University, Washington, DC 20037, USA; dmendel@gwu.edu

**Keywords:** serotonin, dorsal raphe nucleus, median raphe nucleus, sleep-disordered breathing, obstructive sleep apnea, pharmacotherapy

## Abstract

Serotonin or 5-hydroxytryptamine (5-HT) is a ubiquitous neuro-modulator–transmitter that acts in the central nervous system, playing a major role in the control of breathing and other physiological functions. The midbrain, pons, and medulla regions contain several serotonergic nuclei with distinct physiological roles, including regulating the hypercapnic ventilatory response, upper airway patency, and sleep–wake states. Obesity is a major risk factor in the development of sleep-disordered breathing (SDB), such as obstructive sleep apnea (OSA), recurrent closure of the upper airway during sleep, and obesity hypoventilation syndrome (OHS), a condition characterized by daytime hypercapnia and hypoventilation during sleep. Approximately 936 million adults have OSA, and 32 million have OHS worldwide. 5-HT acts on 5-HT receptor subtypes that modulate neural control of breathing and upper airway patency. This article reviews the role of 5-HT in SDB and the current advances in 5-HT-targeted treatments for SDB.

## 1. Introduction

Obesity is a major risk factor in the development of sleep-disordered breathing (SDB). Obesity leads to SDB through multiple mechanisms. Firstly, the accumulation of excess body fat imposes a mechanical load on the respiratory system by physically compressing the upper airway, chest, and lungs [1]. Furthermore, biological responses to the adipocyte-produced hormone leptin, which is integral in appetite regulation and control of breathing, are altered in obesity, leading to leptin resistance [2,3,4,5]. SDB has been linked to insulin resistance [6]. Obesity leads to SDB through multiple mechanisms. Firstly, the accumulation of excess body fat imposes a mechanical load on the respiratory system by physically compressing the upper airway, chest, and lungs [7]. Adipose tissue can produce adipokines such as resistin that contribute to insulin resistance, leading to the development of type-2 diabetes [8]. Obesity can also lead to the infiltration of adipose tissue caused by immune cells and the release of pro-inflammatory molecules such as tumor necrosis factor-ɑ and interleukin-6, increasing the inflammatory response [8,9], which may also contribute to the pathogenesis of SDB [10]. In addition, obesity may disrupt mechanisms by which the central nervous system regulates breathing [11,12]. Breathing is generated by a complex neural network mainly located in the brainstem. The neural network that generates the respiratory rhythm and pattern is responsible for providing rhythmic stimuli to the motoneurons in the spinal cord to control the contraction and relaxation of the respiratory muscles, such as the diaphragm and external intercostals [13,14]. Changes in the neural network that controls breathing may lead to respiratory irregularities in SDB [15].

The most common type of SDB is obstructive sleep apnea (OSA). OSA is characterized by a recurrent closure of the upper airway caused by the loss of muscular tone in the upper airway muscles during sleep [7], which results in recurrent inspiratory flow limitation, termed hypopneas, or the complete cessation of flow, i.e., obstructive apneas. OSA leads to intermittent hypoxemia, hypercapnia, and sleep fragmentation. OSA patients complain of sleepiness, fatigue, and memory loss. In adults, OSA is generally defined by an apnea–hypopnea index (AHI) of five or more events per hour of sleep [16,17], with moderate to severe OSA, defined as an AHI of fifteen or greater. Globally, it is estimated that 936 million adults (30–69 years old) have OSA, and 425 million adults (30–69 years old) have moderate to severe OSA [18]. OSA is two to three times more common in men than women [17]. Another risk factor in the increased prevalence of OSA is a low socioeconomic status, which increases the odds of having OSA by 3.4-fold [19]. The prevalence of OSA increases linearly with age, with approximately 50% of healthy elderly adults over 65 years of age having mild to severe OSA and 20% having mild to moderate OSA [20]. The prevalence of OSA in children ranges from 1% to 4%, but obesity increases it from 5.7% to 56% [21,22]. In children, especially those from 2 to 8 years old, large lymphoid tissues in the upper airway and enlarged tonsils are also seen as risk factors for moderate to severe OSA [23,24]. In all at-risk populations for OSA, obesity has been found to be the leading risk factor. The prevalence of OSA in obese individuals is >50%, and 70% of patients with OSA are obese [25,26]. In total, 10–20% of obese patients with OSA are also diagnosed with obesity hypoventilation syndrome (OHS) [27,28].

OHS is a prevalent and life-threatening severe form of SDB characterized by daytime hypercapnia and hypoventilation during sleep. OHS is driven by decreased CO_2_ sensitivity and depressed ventilatory responses to CO_2_ [27,29,30,31]. The prevalence of OHS in the community has not been determined, but it is estimated at 0.4% (or 1 out of every 260) of the overall global adult population [32]. Increased BMI is associated with the increased prevalence of OHS. The prevalence of OHS is 11% for patients with a BMI of 30–35 (obesity class I), 20% for a BMI of 35–40 (obesity class II), 30% for a BMI of >40 (obesity class III), and 48% for a BMI > 50 kg/m^2^ [33,34]. OHS mortality was reported at 24% during an 18-month observation period [28]. In total, 90% of patients with OHS have co-morbid OSA [35], and an elevation in arterial PaCO_2_ levels can be observed because of the cessation of ventilation during apneic events [35]. In the remaining 10% of patients, OHS is attributed to impaired hypercapnic ventilatory control and sleep-related hypoventilation [36]. Healthy individuals can regulate PaCO_2_ levels by enhancing alveolar ventilation and facilitating CO_2_ removal [37]. However, this compensatory mechanism is disrupted in OHS patients, resulting in CO_2_ retention [38]. During transient hypercapnia, the renal system reduces bicarbonate clearance to counterbalance the acidic pH resulting from elevated CO_2_ [39]. The accumulation of bicarbonate eventually dampens the body’s response to CO_2_, leading to the onset of nocturnal hypoventilation [40].

Currently, the most effective treatment for OSA and OHS is continuous positive airway pressure (CPAP) therapy and non-invasive ventilation (bilevel positive airway pressure, BiPAP). CPAP opens the upper airway with continuous positive pressure through a mask that seals a patient’s nose or both nose and mouth [41,42], whereas BiPAP also inflates the lungs through an inspiratory and expiratory pressure differential. CPAP and BiPAP are equally efficacious treatments for OHS [43,44], but adherence is low at 50–75% because of several concerns, such as mask discomfort, pressure intolerance, sore eyes, costs, and disturbance to partners [45,46,47,48]. One alternative to CPAP is an oral appliance known as a mandibular advancement splint (MAS) [49]. However, compared with CPAP, MASs are less effective in reducing the AHI even though patients adhere to it more than CPAP [50]. The side effects of a MAS, such as pain in the jaw muscles, tooth pain, vomiting, and dry mouth, have made this therapeutic lackluster [51]. In addition, temporal mandibular joint disorder, periodontal disease, and bruxism are contraindications for MASs [52]. Another alternative to CPAP is nasal, palatal, and tongue-based surgeries [53]. Surgical treatments, on average, have been shown to have a 50–70% success rate; however, postoperative management is still required with the potential of worsening or the recurrence of OSA [42,53]. For patients who cannot tolerate CPAP, hypoglossal nerve stimulation is an increasingly accepted preferred alternative. The FDA-approved Inspire device uses an electrical stimulation cuff around the hypoglossal nerve’s distal branch [54]. Contraction of the genioglossus (GG) muscle through hypoglossal nerve stimulation reduced the AHI by 68% and oxygen desaturation by 70%, and efficacy was stable over 3 years during follow-ups, but patients with complete or complex obstructions may not respond well to this treatment [54,55,56,57]. Despite strict exclusion criteria, 33% of participants did not meet the endpoint of a 50% reduction in AHI from baseline and AHI < 20/h [58]. Untreated OSA is associated with increased comorbidities such as stroke, cardiovascular diseases, respiratory diseases, oncologic diseases, metabolic disorders, insomnia, anxiety, and depression [59]. OSA has high all-cause mortality [60], which is decreased by CPAP treatment [60]. There is no FDA-approved pharmacotherapy for OSA or OHS. Novel pharmacological approaches are desperately needed.

## 2. Serotonin and SDB

The pathogenesis of OSA and OHS is driven primarily by altered neural control of breathing and upper airway patency during sleep [32,61]. One of the neurotransmitters that has been implicated in the pathogenesis of OSA is serotonin or 5-hydroxytryptamine (5-HT). 5-HT is a monoamine neurotransmitter formed from the hydroxylation of the essential amino acid L-tryptophan via L-tryptophan hydroxylase (TPH), a rate-limiting enzyme that exists in two isoforms, TPH1 and TPH2, followed by the natural decarboxylation of 5-hydroxytryptophan [62]. TPH1 is highly expressed in the periphery, with the highest expression in the enterochromaffin cells of the intestine [63]. TPH2 controls neuronal 5-HT biosynthesis [64]. The discovery of 5-HT began in the early 1930s with Dr. Vittorio Erspamer, who found an acetone isolate of cells that caused smooth muscle contraction in the enterochromaffin cells in the guts of rabbits, which he named enteramine [65]. Later, the same substance was characterized in the 1940s by Dr. Maurice Rapport, Dr. Irvine Page, and Dr. Green when they isolated a serum vasoconstrictor while researching hypertension, which they named 5-HT [66]. The majority of 5-HT is synthesized in the intestine, but the discovery of 5-HT synthesis in the mammalian central nervous system proved to be important for many behavioral and physiological functions [67]. 5-HT plays a key role in the regulation of mood [68], cardiac function [69], temperature regulation, sleep [70], metabolism [71,72], the control of breathing [73], and upper airway function [74].

5-HT is almost exclusively produced in the neurons of the raphe nuclei of the brainstem and released throughout the central nervous system, where it acts on a range of receptor subtypes to modulate various physiological processes [75]. The effects of 5-HT are terminated when it is taken up into the presynaptic neuron through the 5-HT transporter (SERT) or degraded into 5-hydroxyindolacetic acid (5-HIAA) by monoamine oxidase A (MAOA) (Figure 1) [76]. With 5-HT acting as a master regulator of numerous physiological functions, molecular differences have been found that show that specialized subtypes have unique compartmentalized roles. There are seven types of 5-HT receptors (5-HT1 to 5-HT7) and fourteen subtypes, which are all G-protein-coupled receptors, with the exception of 5-HT3 receptors, which are ligand-gated ion channels. The 5-HT1A, 5-HT2A, and 5-HT2B receptors have been implicated in respiratory control [77,78]. These receptors are found in various regions of the brainstem that are involved in the control of breathing, including the medulla and the pons. 5-HT1A receptors are expressed in the soma and dendrites of 5-HT neurons in the raphe nuclei and have an inhibitory effect on breathing [79]. 5-HT2A and 5-HT2B receptors have been shown to have an excitatory effect on breathing [78].

Clinical research has shown that plasma 5-HT levels are increased in individuals with OSA following CPAP treatment, but the relevance of the findings for central serotonin effects is uncertain [80]. Changes in 5-HT signaling may contribute to metabolic disturbances that are commonly observed in individuals with OSA [81]. For example, the expression of 5-HT2C receptors on pro-opiomelanocortin neurons is associated with the suppression of food intake, the termination of feeding behavior, and decreased body weight [81]. A recent study documented a negative correlation between blood 5-HT levels, oxygen desaturation index, central apnea, and obstructive apnea [82].

## 3. 5-HT Regulation: Neural Control of Breathing and Chemosensitivity in OSA

Respiratory automatism is regulated by the respiratory network located in the brainstem, specifically, in the medulla and pons. The primary respiratory center resides in the medulla, which is responsible for the generation of rhythmic breathing patterns through the interaction of several regions of the ventral respiratory group, such as the Bötzinger complex (BötC) and the pre-Bötzinger complex (pre-BötC) [83,84]. The pre-BötC is located in the ventral medulla and contains inspiratory neurons that display rhythmic activity in phase with inspiration and that are silent during expiration, while expiratory neurons are found in the BötC [13,85,86]. The pontine respiratory groups, parabrachial complex, and Kölliker–Fuse nucleus are also involved in the control of the time of inspiration and in the regulation of the post-inspiratory phase of the respiratory cycle [87,88]. The respiratory rhythm and pattern orchestrated by these centers are greatly influenced by sensory inputs from the central chemoreceptors such as changes in arterial CO_2_ levels, arterial O_2_, and hydrogen ion concentrations [89].

Several areas of the brainstem are implicated in chemoreception. The parafacial region contains the retrotrapezoid/parafacial respiratory group (RTN/pFRG) [68,90]. These neurons are well known for their ability to detect small increases in the partial pressure of CO_2_ (PaCO_2_) in the plasma, as well as the consequent reduction in pH, generating an increase in respiratory activity [91,92,93,94,95]. This respiratory network is also modulated by the peripheral chemoreceptors located in carotid bodies, which detect changes in the partial pressure of oxygen (PaO_2_) PaCO_2_. Besides RTN and carotid bodies, 5-HT cells are found exclusively in the brain cell groups classified as the raphe, which plays a crucial role in the control of breathing [74].

In patients with OSA and OHS, central chemoreceptor sensitivity is controversial. Central chemoreflex may be reduced, which can contribute to the reduced central respiratory drive, which leads to increased occurrences of apneas and hypopneas and decreased hypercapnic ventilatory response [96]. Cheyne–Stokes respiration is a ventilation pattern marked by a gradual increase in tidal volume, followed by a gradual decrease, leading to hyperpnea and apneas (a crescendo–decrescendo pattern). Cheyne–Stokes respiration is most common in congestive heart failure with decreased left ventricular ejection fraction, but it can also be seen in healthy subjects at high altitudes [97]. Although augmented chemoreceptor sensitivity has been reported in patients with Cheyne–Stokes respiration and subsets of OSA patients, patients with OHS have reduced peripheral chemoreceptor sensitivity, which is also reduced in patients with OSA, leading to blunted hypoxic ventilatory responses and hypercapnic ventilatory responses [98]. However, in subsets of patients, such as premenopausal women and heart failure patients with increased hypopneas, chemoreceptor sensitivity is increased [99,100]. This suggests that there are different phenotypes involving central chemoreceptor sensitivity in SDB.

## 4. Anatomy and Function of the Raphe Nuclei

Physiological roles can be found in the pontomedullary area. 5-HT projections that modulate the neural regulation of breathing have been shown to originate in the raphe nuclei of the brainstem, pons, and midbrain. From the caudal to rostral regions, the raphe nuclei include the nucleus raphe obscurus (RO) and nucleus raphe pallidus (RP) in the medulla, the nucleus raphe magnus (RM), the nucleus raphe pontine (RPo), and the median raphe nucleus (MR) in the pons, along with the dorsal raphe nucleus (DR) situated in the midbrain (Figure 2).

The mechanisms by which 5-HT-raphe neurons regulate breathing remain unknown. Dense populations of 5-HT-containing axons are found in the rostral to caudal portion of the respiratory network [101]. Immunofluorescence labeling has revealed an increase in the number of 5-HT nerve fibers during hypoxia compared with controls in the rostral ventrolateral medulla, the caudal ventrolateral medulla, the lateral part of the nucleus of the solitary tract, and the dorsal motor nucleus of the vagus nerve in rats [102]. A detailed distribution of serotonergic projections has been found to be rostral to caudal of the BötC, the pre-BötC, the anterior rostral ventral respiratory group, and the posterior rostral/caudal ventral respiratory group [103]. In addition, neurons labeled with the cholera toxin B subunit and the TPH2 enzyme have mainly been found in the caudal raphe nuclei and partly in the rostral raphe nuclei. The findings of Morinaga et al. suggest that serotonergic projections originate not only in the caudal portion of the raphe but also in the rostral portion and project to several locations of the ventral respiratory column [103].

RO, a midline structure that runs the length of the medulla oblongata, is especially important for ventilation, hypercapnic ventilatory response, heart rate, blood pressure, movement, coordinated locomotion, and muscle tone [104]. The photostimulation of RO neurons increases respiratory frequency and diaphragm EMG amplitude, but during hypercapnia, there is no response, suggesting that they are not involved in central chemosensitivity [105]. The selective stimulation of the 5-HT neurons in the RO stimulates breathing, whereas the blockade of 5-HT2 receptors attenuates breathing [105]. It has been shown that, during hypercapnia, the 5-HT2 antagonist decreases the hypercapnic ventilatory response more in treated rats compared with untreated rats [106]. In one study, genetic knockout mice with a 5-HT precursor that nearly abolishes central 5-HT neurons had a 50% decrease in hypercapnic ventilatory response compared to controls [107]. However, no changes were found in the baseline ventilation and hypoxic ventilatory responses of knockout mice compared with controls until the intracerebroventricular infusion of 5-HT reversed the blunted hypercapnic ventilatory response, and the baseline ventilation was increased.

RM primarily innervates the dorsal horn of the spinal cord and appears to be primarily implicated in sensory control [108]. During hypercapnia, 5-HT neurons of the RM are stimulated. The disruption of RM innervation blunts HCVR [109,110]. Therefore, the RM plays a role in central chemosensitivity during hypercapnia. During acute and chronic intermittent hypoxic conditions, 5-HT1A receptor activation decreases the hypoxic ventilatory response, and 5-HT expression in the RMR (rostral medullary raphe) is reduced [111]. In one study, the 5-HT1A activation of the medullary raphe also decreased the hypercapnic ventilatory response in unanesthetized rats during wakefulness and NREM sleep; lesions in the RM housing 5-HT neurons increased the hypoxic ventilatory response in the rats, but no change in ventilation occurred during normoxic conditions [112]. The authors suggested that the RM specifically exerts inhibitory modulation on the control of breathing via tidal volume but does not modulate the control of breathing in resting conditions. Andrzejewski et al. found that the intraperitoneal administration of the 5-HT2 receptor agonist during normoxic conditions increased minute ventilation and the respiratory rate. Then, the subsequent intraperitoneal administration of the 5-HT2 receptor antagonist reverted minute ventilation and the respiratory rate to control values [113]. When hypoxia was induced after 5-HT2 receptor activation, the hypoxic ventilatory response decreased, which correlated with a decrease in the 5-HT levels of the brainstem and striatum via double chemical injections of 6-hydroxydopamine that targeted the dopaminergic and serotonergic neurotransmission system.

RP and the parapyramidal region (B6 group) primarily target the intermediolateral cell column and ventral horn and are involved in the autonomic control of body temperature [114]. The RP also innervates neurons in the spinal cord and brainstem involved in cardiovascular function. In one study, the injection of 6-hydroxydopamine into rodents to model Parkinson’s disease showed decreased RP neurons that projected to the RTN but no changes in the number of hypercapnic activated neurons in the raphe [115]. This suggests that the RP is not involved in central chemosensitivity during hypercapnia.

The DR is the largest serotoninergic nucleus located in the midbrain, primarily meant to send projections to the limbic system and the cortex. 5-HT neurons in the DR are also active during wakefulness and become less active during NREM sleep [82,116]. CO_2_ stimulation of 5-HT neurons in the DR leads to arousals from sleep independent of respiratory activation [117]. In one study, the genetic deletion of 5-HT neurons in the DR and the optogenetic silencing of DR 5-HT neuronal projections to the terminals in the parabrachial nucleus (PBel) increased arousals during hypercapnia [118]. Furthermore, when a 5-HT2a receptor agonist was administered while 5-HT neurons of the DR were inhibited in the PBel, the arousals were restored during hypercapnia.

Given existing evidence showing the role of 5-HT in the neural control of breathing and central chemosensitivity, our lab examined the effects of brain 5-HT deficiency on the neural control of breathing during sleep, the hypercapnic ventilatory response, and CO_2_ production in young and old mice. We found that, regardless of age, brain-5-HT-deficient mice exhibited suppressed control of breathing during NREM and REM sleep. During wakefulness, 16–18-week-old mice with brain 5-HT deficiency maintained ventilation and hypercapnic ventilatory response during hypercapnia, whereas older 40–44-week-old 5-HT-deficient mice had decreased ventilation and hypercapnic ventilatory responses [119]. These data suggest that 5-HT plays a role in the neural control of breathing during sleep, which is magnified with aging. 5-HT activates RTN neurons, which are intrinsic respiratory chemosensors, through 5-HT7 receptors, suggesting the involvement of 5-HT7 receptors in chemosensitivity in vitro [90]; however, more recent studies have shown that the role of the 5-HT7 receptor in RTN responses in vivo is minimal [120].

In OSA, the elevation of CO_2_ levels not only affects the neural control of breathing and upper airway patency but also contributes to awakening. As already mentioned, experimental evidence shows that serotonergic input from the DR to the PBel via 5-HT2a receptors is critical for hypercapnic-induced arousal [118]. The eventual therapeutic approach to OSA and OHS may involve increasing breathing and the improvement of upper airway patency without increasing the arousal threshold. To achieve this goal, the mechanisms involved in the arousals and the role of 5-HT released by the dorsal raphe must be completely understood.

## 5. 5-HT Regulation: Upper Airway Function in OSA

In addition to its role in the neural control of breathing, 5-HT has also been implicated in the regulation of upper airway motor control. During eupneic breathing, the upper airway is kept open by the activity of the muscles of the tongue, soft palate, and pharynx [121]. However, in people and animal models of OSA, the upper airway becomes narrowed, and a total or partial collapse during sleep can occur, leading to hypopneas, apneas, repeated arousals, and impaired ventilatory responses to hypercapnia and hypoxia [122]. The VOTE classification is commonly used to identify the degree and configuration of obstruction related to the velum, oropharyngeal lateral walls, tongue base, and epiglottis [123].

5-HT has been shown to modulate airway muscle tone [74] via serotonergic neurons in the medullary raphe that provide excitatory input to hypoglossal motoneurons [121]. The largest upper airway dilator muscle, the GG, is involved in the patency of the upper airway through reflexive activation. In OSA patients, there is a loss of GG muscle activity, especially during REM sleep, that contributes to the narrowing and collapse of the upper airway [124]. REM sleep is associated with greater sympathetic activity and cardiovascular instability compared with NREM sleep in OSA patients [125]. It has also been found that, in REM sleep, obstructive apneas and hypopneas increase in duration and frequency with greater oxygen desaturation compared with NREM sleep. However, during wakefulness, OSA patients have exhibited increased GG muscle activity, which suggests a neuromuscular compensatory mechanism [126]. Serotoninergic drugs have been shown to be effective in the treatment of OSA in animals, including the English bulldog, using serotonin antagonists and reuptake inhibitors (such as trazodone), serotonin precursors, and 5-HT3 blockers (such as ondansetron) [127,128]. Jelev et al. showed that 5-HT caused marked GG activation during NREM and REM sleep in rats [129]. However, most of the studies evaluating the role of 5-HT in upper airway patency did not measure breathing during sleep and did not evaluate upper airway obstruction.

In one study, the selective lesioning of serotonergic axonal terminals through a neurotoxic injection increased the frequency and duration of apneas during NREM sleep, but no changes were observed in REM sleep [130]. Tph2 knockout mice (Tph2^−/−^) displayed increased central apnea frequency and duration, decreased apnea events coupled to arousals, and decreased ventilatory responses to hypercapnia, as well as hypoxia during NREM sleep. During REM sleep, Tph2^−/−^ mice displayed similar apnea frequency and duration to Tph2^+/+^ mice. When Tph2^+/+^ mice were compared between NREM and REM sleep, they displayed greater apnea duration in REM sleep [131].

Normally, the phrenic nerve stimulates the diaphragm to contract and initiate inhalation, but in OSA patients, increased respiratory effort is required because of an airway obstruction, even with intact phrenic nerve function, leading to arousal [132]. Acute intermittent hypoxia-induced phrenic nerve activity is thought to produce stable breathing that minimizes apneas and hypopneas in patients with OSA. In one study, when 5-HT1A receptors are blocked through a selective antagonist injection to the raphe nuclei in Sprague–Dawley rats exposed to acute intermittent hypoxia, the induction of phrenic nerve activity was prevented [133].

The administration of a selective 5-HT1A agonist to a mouse model of Rett syndrome, which is known to lead to SDB in patients, decreased apneas in [134,135]. 5-HT2A receptor blockade through an antagonist reduced apneas during NREM sleep, but apneas were unchanged for 5-HT2A knockout mice during NREM sleep, which suggests an alternative unknown adaptive mechanism [136]. In one study, the administration of 5-HT2A and 5HT2C receptor antagonists in older obese Zucker rats decreased minute ventilation and the respiratory rate, prolonged inspiratory time, and increased oxygen consumption. Pharyngeal critical pressure was also greater in older obese rats compared with older lean rats, but it was increased in both older obese and lean mice with the administration of the 5-HT2A and 2C receptor antagonists, indicating a more collapsible upper airway. Based on these findings, the researchers suggested that older obese rats specifically may utilize serotonergic control, acting on 5-HT2A and 2C receptors to control upper airway dilator muscles as a mechanism to prevent upper airway collapse [137]. The injection of a 5-HT3 receptor antagonist with 5-HT4 receptor agonist properties decreased central apneas by 50% during NREM sleep and by 80% during REM sleep [138].

Given the current evidence, 5-HT receptors play a significant role in regulating upper airway function, particularly in the pathogenesis of obstructive sleep apnea (OSA). The activation of 5-HT1A receptors in the upper airway muscles can solve upper airway collapsibility and obstructions, which improves the severity of OSA [139]. Additionally, the antagonistic targeting of 5-HT receptor subtypes, such as 5-HT2A, 5-HT2C, and 5-HT3, may offer a potential therapeutic approach to the treatment of OSA. However, further research is needed to fully understand the sites of action and mechanisms underlying the effects of 5-HT receptors on upper airway function and to develop more effective and targeted treatments for OSA. In addition, most investigations have been performed on animals; thus, more studies on human OSA patients need to be conducted.

## 6. Clinical Implications of 5-HT Studies in Treating SDB

Animal studies targeting 5-HT have shown promising results in treating SDB. However, numerous clinical studies focused on using pharmacotherapy to target 5-HT receptors to treat OSA have been met with limited success. Clinical studies have tested serotonin reuptake inhibitors (SRIs) to treat OSA. In one study, the administration of the SRIs fluoxetine and protriptyline together showed an overall decrease in baseline AHI by 40% during NREM sleep, but no changes in AHI during REM sleep were observed [140]. AHI was measured via overnight polysomnography, but individual data were widely variable, with only half of the patients showing a response, and REM sleep time was markedly reduced, making it difficult to assess [140]. Paroxetine elevated peak GG muscle activity during NREM sleep, yet it did not induce any changes in the AHI or sleep architecture, except for a reduction in REM sleep duration while on paroxetine [141]. However, a later study with more patients showed an improvement in AHI by 18% compared with a placebo specifically during NREM but not in REM sleep; however, it was not a pronounced change. In addition to SRIs, specific 5-HT receptor agonists and antagonists have been examined in clinical studies. Ondansetron, a 5-HT3 antagonist and antiemetic, widely used in everyday practice, showed no effect on OSA patients in a single-night crossover design [142]. One clinical study with buspirone, a 5-HT1A agonist, reduced AHI by 36% compared with a placebo, but this was in four out of five patients [143]. With a lack of specificity, it is unclear if many of the drugs that target 5-HT receptors have effects on other 5-HT receptor subtypes or diverse populations of neurons. In addition, given the heterogeneity of the patient population that has OSA and severity, it is difficult to identify why some patients respond well while others fail to do so. Some drugs are also known to have adverse effects, such as on sleep quality, dry mouth, urinary hesitancy, mild constipation, and erectile dysfunction, which adds to the implications of drugs that target 5-HT [144,145]. Despite these challenges, it is important to note that these studies have provided insight into the underlying mechanisms and possible therapeutics for OSA.

## 7. Conclusions

In conclusion, further research on the specific roles and mechanisms of 5-HT receptor subtypes should be explored to develop a pharmacotherapy that treats OSA. The efficacy of 5-HT agonists or SRIs on OHS and OSA depends on the patient’s phenotype. Simple phenotyping tools based on regular sleep studies, a greater understanding of the 5-HT receptor subtypes, and 5-HT action at different sites within the respiratory circuitry are needed to reassess the role of 5-HT modulation in SDB treatment. There are limitations on current 5-HT pharmacotherapies in clinical studies given the low number of human studies compared with animal studies, the low number of participants, and the increased variability in the outcomes of treatments.

## Figures and Tables

**Figure 1 ijms-25-01483-f001:**
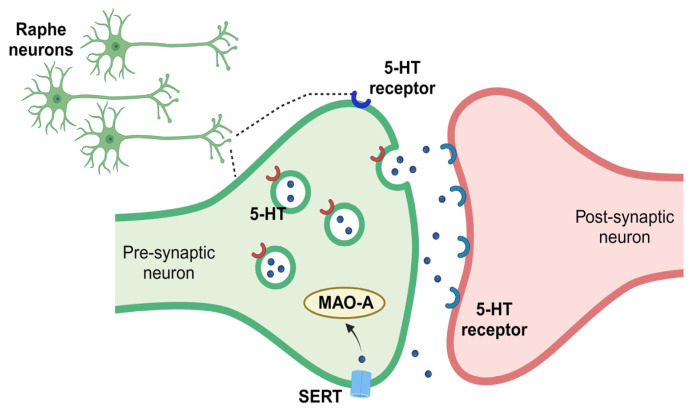
Simplified 5-HT release and metabolism. 5-HT is synthesized by pre-synaptic raphe neurons. After synthesis, 5-HT is transported into vesicles for storage. Upon vesicle fusion with the plasma membrane, 5-HT is released, where it interacts with 5-HT autoreceptors located on the releasing cell or 5-HT heteroreceptors, located on post-synaptic neurons of other regions. The serotonin transporter (SERT) transports 5-HT back into the releasing cell, where it is likely to be repackaged for release by or degraded by monoamine oxidase A (MAO-A).

**Figure 2 ijms-25-01483-f002:**
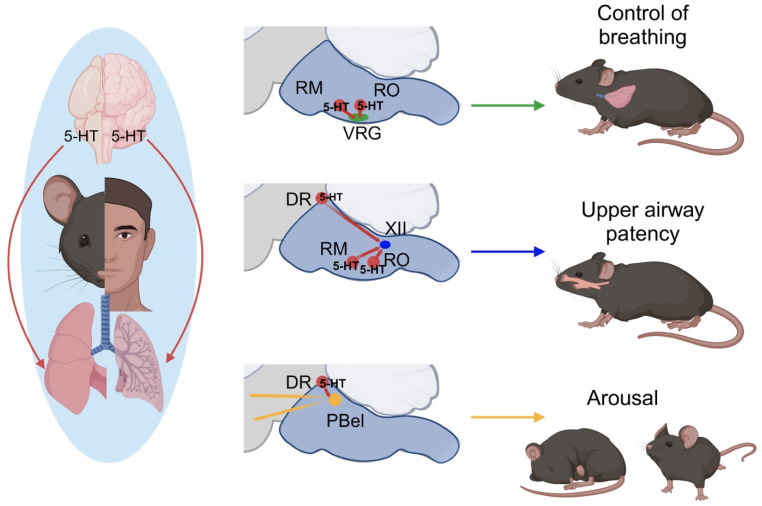
Schematic representation of the location of 5-HT neurons within the raphe nuclei involved in the modulation of 5-HT projections associated with sleep-disordered breathing. Clusters of 5-HT neurons found in the nucleus raphe magnus (RM), nucleus raphe obscurus (RO), and ventral respiratory group (VRG) modulate the neural control of breathing. 5-HT neurons in the dorsal raphe nucleus (DR), RM, and RO project to the hypoglossal nerve (XII) to modulate upper airway patency. 5-HT neurons in the DR project to the parabrachial subnucleus (PBel) to modulate arousals.

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
