# Peer review of "Revisiting the Role of Serotonin in Sleep-Disordered Breathing"

_ijms, 2024, doi:10.3390/ijms25031483_

Round 1

Reviewer 1 Report

Comments and Suggestions for Authors

I reviewed the manuscript titled Revisiting the role of serotonin in sleep-disordered breathing.The paper is well written and is of interest to sleep medicine researchers. A few minor issues:

1)Abstract 16 Sleep-disordered breathing encompasses a broad spectrum of sleep-related breathing disorders, including obstructive sleep apnea (OSA), central sleep apnea, as well as sleep-related hypoventilation and hypoxemia, which is why OSA is classified as SBD. Are these numbers correct? Confusing if there are more OSA than SBD.

2) Introduction 24 This appears to be an OSA definition, not an SBD definition. Please do not confuse these concepts.

Reviewer 2 Report

Comments and Suggestions for Authors

First, I would like to thank you for reading and reviewing this manuscript. This manuscript summarises the role of serotonin in sleep-disordered breathing, which is an interesting topic. However, some issues must be corrected before this manuscript can be considered for publication. Regarding this, see my recommendations in the following.

Abstract

Line 9. When introducing the abbreviation, also include 5-hydroxytryptamine, of which the abbreviation is 5-HT.

Lines 9-10. I also recommend mentioning that serotonin plays a significant role in other physiological functions.

Keywords

I recommend including ‘obstructive sleep apnoea’.

Introduction

Line 25. It would be beneficial to mention that a total airway closure is happening during an apnea. Furthermore, the result of obstructions, i.e., oxygen desaturation, should also be mentioned.

Line 26. Considering obesity as an essential risk factor for sleep-disordered breathing, I recommend including the following citations:

[Povitz M, James MT, Pendharkar SR, Raneri J, Hanly PJ, Tsai WH. Prevalence of Sleep-disordered Breathing in Obese Patients with Chronic Hypoxemia. A Cross-Sectional Study. Ann Am Thorac Soc. 2015 Jun;12(6):921-7. doi: 10.1513/AnnalsATS.201412-551OC.] 

[Molnár V, Lakner Z, Molnár A, Tárnoki DL, Tárnoki ÁD, Kunos L, Tamás L. The Predictive Role of Subcutaneous Adipose Tissue in the Pathogenesis of Obstructive Sleep ApnoeaLife. 2022;12:1504. doi: 10.3390/life12101504]

Furthermore, in addition to the mechanical effects of the adipose tissue, its hormonal and inflammatory effects should also be discussed.

Line 37. From my point of view, ‘upper airway’ muscles would be a better term in this case, as not only the pharyngeal muscles are responsible for obstructions.

Lines 43-48. Regarding the epidemiologic features of OSA, its prevalence in children should also be mentioned; in this case, besides obesity, the size of upper airway lymphoid tissues is an essential risk factor. Please discuss this and cite accordingly:

[Kanney ML, Harford KL, Raol N, Leu RM. Obstructive sleep apnea in pediatric obesity and the effects of sleeve gastrectomy. Semin Pediatr Surg. 2020;29(1):150887. doi: 10.1016/j.sempedsurg.2020.150887]

[Dékány L, Molnár V, Molnár A, Bikov A, Lázár Z, Bárdos-Csenteri O, Benedek P. Analysis of possible risk factors for the severity of paediatric obstructive sleep apnoea syndrome. Eur Arch Otorhinolaryngol. 2023 Dec;280(12):5607-5614. doi: 10.1007/s00405-023-08237-w.]

Lines 59-60. I would mention what these BMI categories refer to (i.e., obesity class I, etc.).

Line 62. I recommend specifying that an elevation in arterial PaCO2 levels can be observed.

Line 81. Considering MAS, I recommend including its relative contraindications, e.g., TM joint disorders, periodontal disease, etc.

Lines 82-83. Some examples regarding the surgical treatment options for upper airway obstruction need to be included. E.g., in addition to palatal and tongue-base surgeries, tonsil and pharynx surgeries are also essential, such as tonsillectomy/tonsillotomy or pharyngeal reconstructions. Please clarify this. Furthermore, the risk of peri-and postoperative complications must also be mentioned. In addition, contraindications to general anaesthesia can limit the use of surgical treatment, especially in comorbid patients.

Lines 117-118. In addition to the currently included examples, I recommend mentioning the importance of serotonin in the central auditory structures and its related symptoms, including the following reference articles:

[Kaltenbach JA. The dorsal cochlear nucleus as a participant in the auditory, attentional and emotional components of tinnitus. Hear Res. 2006 Jun-Jul;216-217:224-34. doi: 10.1016/j.heares.2006.01.002. Epub 2006 Feb 15]

[Molnár A, Mavrogeni P, Tamás L, Maihoub S. Correlation Between Tinnitus Handicap and Depression and Anxiety Scores. Ear Nose Throat J. 2022 Nov 8:1455613221139211. doi: 10.1177/01455613221139211]

Lines 121-124. A figure presenting these mechanisms would significantly improve the quality of the manuscript.

Lines 135-136. These are promising results; however, from my point of view, it should be mentioned that detecting serotonin plasma levels is a limitation concerning central serotonin effects.

Line 139. It would be essential to detail these regulation effects on appetite and energy expenditure. 

Lines 143-144. In this case, the reticular formation should be mentioned.

Lines 153-154. Regarding these chemosensory receptors, more details are necessary. E.g., what is their physiological trigger, etc?

Line 168. It would be beneficial to mention pathologies in which a Cheyne-Stokes breathing pattern can be detected and what this respiration pattern practically means.

Figure 1. An explanation of why rats and humans are also presented in this Figure is missing.

Line 192. This increasing tendency must be specified, i.e., the size/diameter was increased, etc.

Line 204. Could the authors mention some examples regarding these outputs?

Lines 204-205. More specifically, an increased respiratory frequency and diaphragmatic EMG amplitude were observed in that study.

Line 221. Previously, the ‘RM’ abbreviation referred to the nucleus raphe magnus; however, here, it is explained as a rostral medullary raphe, which can be quite confusing.

Line 228. Explain the abbreviation of ‘IP’.

Line 233. Can the authors specify this neurotoxic compound?

Line 238. Explain the ‘OHDA’ abbreviation.

Lines 251-260. In the case of this section, it must be clarified that the authors are presenting their previous results. Furthermore, even if these results are the authors’ results, the reference article must be included. 

Line 278. I recommend specifying that a total or partial collapse during sleep can occur. Furthermore, in everyday practice, in people, the location of the obstruction can be categorised based on the VOTE classification. Please mention this.

[Molnár V, Molnár A, Lakner Z, Tárnoki DL, Tárnoki ÁD, Jokkel Z, Kunos L, Tamás L. The prognostic role of ultrasound and magnetic resonance imaging in obstructive sleep apnoea based on lateral oropharyngeal wall obstruction. Sleep Breath. 2023 Mar;27(1):319-328. doi: 10.1007/s11325-022-02597-z]

Line 304. I recommend clarifying that the phrenic nerve innervates the diaphragm, and its role in normal breathing and OSA.

Line 310. Always use the abbreviation of sleep-disordered breathing after its first appearance in the text.

Line 327. First, it would be more accurate to mention that upper airway collapsibility and obstructions are solved, which improves the severity of OSA.

Line 328. In this case, the importance of these specific receptors in OSA treatment should be highlighted.

Lines 330-332. It must also be mentioned that most investigations were performed on animals, not human OSA patients.

Line 336. In this case, serotonin reuptake inhibitor would be a better term.

Line 338. Considering this investigation, only the effects in NREM sleep are presented. What has happened during REM sleep? Furthermore, the method measuring the AHI should also be mentioned (i.e., overnight polysomnography or portable polygraphy). 

Line 341. What has happened, considering the other sleep test parameters?

Line 342. In this case, it must be mentioned that an 18% change in the AHI is not a pronounced change.

Line 344. The accurate spelling is ondansetron; please correct it. Furthermore, I would mention that this medicine, as an antiemetic, is widely used in everyday practice.

Lines 349-350. Regarding this, the severity of OSA may also contribute.

Line 351. I recommend specifying these ‘other aspects’.

At the end of the manuscript, the limitations of the previous studies on 5-HT pharmacotherapy must be summarised. E.g., most studies carried out on animals, human studies involving a low number of participants, etc.

A conclusion/summary section is completely missing, which should be added.

I am looking forward to receiving the revised version of the manuscript, which includes a point-by-point response to each review comment with all required changes accurately made. This is necessary for deciding whether this manuscript can be considered for publication. 

Comments on the Quality of English Language

Only minor corrections in English are necessary. 

Round 2

Reviewer 2 Report

Comments and Suggestions for Authors

I would like to thank the authors for their cooperation; the quality of the manuscript has significantly been improved. However, some minor issues still remain, which must be corrected. 

Concerning my first review comments, the following still need to be included from my point of view. I respectfully accept the authors’ opinion; however, even considering that they are not the topic of this review, other physiological functions of serotonin were included and surgical therapies were mentioned. Consequently, I would like to ask the authors to include the following:

In addition to the currently included examples, I recommend mentioning the importance of serotonin in the central auditory structures and its related symptoms, including the following reference articles: 

[Molnár A, Mavrogeni P, Tamás L, Maihoub S. Correlation Between Tinnitus Handicap and Depression and Anxiety Scores. Ear Nose Throat J. 2022 Nov 8:1455613221139211. doi: 10.1177/01455613221139211]

Some examples regarding the surgical treatment options for upper airway obstruction need to be included. E.g., in addition to palatal and tongue-base surgeries, tonsil and pharynx surgeries are also essential, such as tonsillectomy/tonsillotomy or pharyngeal reconstructions. Please clarify this. Furthermore, the risk of peri-and postoperative complications must also be mentioned. In addition, contraindications to general anaesthesia can limit the use of surgical treatment, especially in comorbid patients.

One additional minor necessary correction:

Line 287. After introducing the abbreviation of genioglossus muscle, it should be used in the whole text. 

Comments on the Quality of English Language

Minor corrections are necessary. 
